# Viremia Kinetics in Pigs Inoculated with Modified Live African Swine Fever Viruses

**DOI:** 10.3390/vaccines13070686

**Published:** 2025-06-26

**Authors:** Alexey Sereda, Mikhail Vlasov, Timofey Sevskikh, Andrey Koltsov, Galina Koltsova

**Affiliations:** Federal Research Center for Virology and Microbiology, Academician Bakoulov Street, bldg. 1, Petushki Area, 601125 Volginsky, Vladimir Region, Russia; vlasovmikhail1993@yandex.ru (M.V.); sefskih@mail.ru (T.S.); kolcov.andrew@gmail.com (A.K.); burmakinags@gmail.com (G.K.)

**Keywords:** African swine fever, live attenuated vaccine, safety and efficacy of vaccines, viremia

## Abstract

Background: African Swine Fever (ASF) is a viral hemorrhagic disease characterized by diverse clinical and pathological manifestations depending on the virulence of isolates/strains and the immunological status of pigs. The use of modified live viruses (MLVs) is currently the most common approach in developing vaccines against ASF. However, despite the availability of dozens of MLV candidates that meet basic safety and efficacy criteria—such as the absence of severe clinical signs and survival after challenge with a virulent strain—no broadly accepted vaccine has yet been developed. Here, we propose viremia testing as an essential criterion for evaluating candidate ASF vaccines, with levels exceeding 10^4^ HAD_50_/TCID_50_ and lasting longer than 21–28 days post vaccination considered unfavorable indicators. Methods: We analyzed ASF MLV vaccines obtained through the deletion of one, two, or more genes, focusing on viremia kinetics after vaccination and challenge with virulent ASFV strains. Post mortem data were used to assess viral persistence in organs. Results: Most MLV candidates, especially those with single-gene deletions, demonstrated relatively high viremia levels after vaccination and challenge. Viral persistence was frequently detected in organs upon necropsy. MLVs with an additional EP402R gene deletion showed low viremia after vaccination but high levels after challenge. Nevertheless, several candidates with favorable viremia profiles were identified, including those obtained via targeted deletions or serial passaging in cell cultures. Conclusions: Incorporating viremia assessment as a primary screening criterion can significantly narrow down the selection of promising MLV candidates and help accelerate the development of effective emergency vaccines for use in ASF-affected regions.

## 1. Introduction

African Swine Fever (ASF) is a hemorrhagic viral disease in swine, with variable clinical outcomes and pathological features influenced by the virulence of the infecting strain or isolate and the host immune response [1,2]. The causative agent, African Swine Fever Virus (ASFV), is the only member of the *Asfarviridae* family and represents the sole known DNA-based arbovirus. This large enveloped virus possesses a double-stranded DNA genome of approximately 190 kilobase pairs that encodes over 170 distinct viral proteins [3,4]. The ASF virus is heterogeneous in pathogenicity, ability to induce hemadsorption, and genome length. To date, ASFV isolates/strains have been classified into 23 genotypes and 9 seroimmunotypes [5,6,7]. Based on virulence, ASFV strains are classified as highly, moderately, and low-virulent [8,9,10,11]. Depending on the virus virulence, the disease manifests in hyperacute, acute, subacute, chronic, and subclinical forms, with mortality ranging from 90–100% to none [11,12]. ASFV causes prolonged persistent infection in warthogs and domestic pigs that survive the acute viral infection [13]. In controlled experimental settings, ASFV demonstrates extended persistence in many domestic pigs infected with moderately virulent isolates, with detectable viral shedding lasting up to 70 days post infection (dpi) [14]. Following either oronasal (ON) or intramuscular (IM) exposure, the virus initially replicates in mononuclear phagocytes located in the tonsils, submandibular lymph nodes, and other regional lymphoid tissues before disseminating via the lymphatic and circulatory systems to secondary replication sites, where it can be identified within 2–3 dpi [15]. Extensive replication and necrosis observed in macrophages in various organs result in a large amount of free virus in the interstitial space [16,17]. Hemadsorption is a characteristic feature of ASF, first discovered in vitro and subsequently used for diagnostic purposes [18]. There is a close relationship between hemadsorption, virus budding, and the presence of viral particles adhering to red blood cells in small invaginations of the cytoplasmic membrane, which is a hallmark of ASFV spread at the earliest stages of the disease [19,20]. The earliest live vaccines, derived through the serial passage of virulent strains in the cell culture to reduce virulence, were associated with 10–50% pig mortality when deployed in Portugal and Spain during the 1960s [21,22]. Despite efforts to develop alternative platforms such as inactivated, subunit, adenovirus-vectored, and DNA vaccines, none have demonstrated sufficient efficacy [23,24,25,26,27]. At present, attenuated strains remain the leading strategy for developing protective vaccines against ASF. Genetically engineered attenuation offers a scientifically grounded and potentially safer option compared to naturally attenuated isolates [28].

In 2021, the World Organisation for Animal Health (WOAH) launched initiatives aimed at establishing standardized evaluation protocols for first-generation, modified, and live ASFV-based vaccines (ASF MLV) [29]. Central safety indicators include the lack of observable ASF-related clinical symptoms and animal survival following both vaccination and subsequent challenge. Additionally, comprehensive safety profiling is advised to cover parameters such as viremia levels, viral excretion patterns, organ pathology, persistence, transmissibility, reversion to virulence, and the genetic stability of the viral genome [29].

Viremia, which reflects viral replication and transmission under natural conditions, is an important criterion for evaluating vaccine safety. Studies on swine behavior show that social interactions, such as feeding or mating, often cause skin damage. The resulting bleeding may lead to more biting and licking [30]. Consequently, blood easily contaminates the environment, especially during intra-pen transmission. Additionally, blood is traditionally considered the primary route of ASFV transmission due to the high viral load in infected animals [31,32]. Accumulated data clearly indicate that animals recovering from ASF infection may still play a role in virus transmission and spread. It cannot be ruled out that the virus may remain latent in some tissues and reactivate under certain natural or induced conditions (transportation, insufficient feeding, immunosuppression, etc.), thereby facilitating transmission.

We analyzed research results evaluating ASF MLV based on viremia levels after vaccination and challenge infection. The analysis primarily relied on the authors’ interpretation of the presented results.

ASF MLV was categorized into two groups: (A) “problematic” and (B) “potential”. Group A further subdivided ASF MLV into three subgroups: those with one deletion; those with two or more deletions; and those with two deletions, one of which involves the EP402R gene. Within each group (subgroup), ASF MLV was ranked from high to low viremia levels. A borderline viremia level was set at 10^4^ HAD_50_ (or TCID_5050_/mL), corresponding to the dose in 1 mL required for oral pig infection [33].

Based on viremia levels on the day of challenge with the virulent virus (21–45 dpi) and the end of the experiments (21–28 days post challenge, dpc), ASF MLV was ranked into three groups: (1) absent; (2) low, ≤10^4^ HAD_50_ or TCID_50_/mL; and (3) high, >10^4^ HAD_50_ or TCID_50_/mL. It was assumed that a virus titer of 10^4^ HAD_50_/mL or TCID_50_/mL corresponded to Ct values of around 32 in real-time PCR or 10^4^ copies of ASFV genome/mL (Ct = 18–24) in qPCR [34,35].

## 2. ASF MLV with a Single Deletion

It has been demonstrated that several attenuated strains obtained through genetic manipulations involving the deletion of a single gene induce protection against the virulent parental virus [36].

### 2.1. ASFV-G-∆H108R

The deletion of the H108R gene from the genome of ASFV-Georgia2007 (ASFV-G) resulted in a significant reduction in virus virulence, except for one animal that developed a slightly delayed fatal form of the disease [37]. The remaining animals (four individuals) infected IM with ASFV-G-∆H108R did not exhibit any clinical signs associated with ASF, except for transient and mild fever spikes. Viremia levels peaked at 11 dpi, reaching 10^7^–10^8^ HAD_50_/mL, and decreased by 28 dpi to 10^4^–10^7^ HAD_50_/mL.

Four pigs that survived vaccination with ASFV-G-∆H108R were challenged IM with 10^2^ HAD_50_ of the parental virus at 28 dpi. The animals remained clinically normal throughout the 21-day observation period, with no transient fever observed. Viremia levels gradually decreased until the end of the experimental period (21 dpc), when titers ranged from undetectable (10^1.8^ HAD_50_/mL) to 10^4.3^ HAD_50_/mL. The post mortem viral loads in the organs and tissues of surviving animals are not presented.

### 2.2. SY18∆I226R

Zhang Y. et al. (2021) reported on an ASF vaccine candidate named SY18∆I226R, which contains a targeted deletion of the I226R gene [38]. This genetic modification leads to full attenuation, as no clinical symptoms were observed in pigs following intramuscular inoculation with either 10^4^ or 10^7^ TCID_50_ of the modified virus. Although viremia was detected and showed a gradual decline over time, viral shedding was only sporadically identified in oral or anal swabs. When challenged intramuscularly with the parental ASFV SY18 strain, all vaccinated animals survived for 21 dpi without showing signs of fever or other clinical manifestations of ASF. No pathological changes were observed in surviving pigs immunized with SY18DI226R. Viral DNA was not detected upon necropsy after viremia disappearance. The qPCR results were negative when testing samples of all tissues of SY18DI226R-immunized pigs in both low- and high-dose groups.

### 2.3. ASFV-G-ΔI177L

Borca M.V. et al. (2020) demonstrated that removing the I177L gene from the highly virulent ASFV-G isolate results in the full attenuation of the virus in pigs [39]. Animals inoculated intramuscularly with the I177L-deleted strain (ASFV-G-ΔI177L) showed no clinical signs and were completely protected against challenge with the virulent parental virus.

Although clinically infected ASFV-G-ΔI177L animals remained normal, viremia was observed by 28 dpi in all animals, with relatively high titers in some cases. As observed in previous studies, the presence of long-term viremia or virus persistence is not a rare event in animals inoculated with other attenuated ASFV strains [40,41]. It was considered to be associated with protection against challenge with virulent ASFV.

After challenge, viremia levels gradually decreased until the end of the experimental period (21 dpi), when the circulating virus was undetectable in the blood of any of these animals. In most pigs across both groups, the vaccine virus was found either in the tonsils or spleen.

The ASFV-G-ΔI177L strain can be delivered via the oronasal route with protective efficacy comparable to that achieved through intramuscular administration. Pigs vaccinated via the oronasal route were fully protected against infection with virulent ASFV-G [42]. Notably, viremia levels were lower in these animals compared to those receiving the vaccine intramuscularly. No clinical signs of ASF were observed in either group throughout the 28-day pre-challenge monitoring period.

At the conclusion of the study (21 dpc), tonsil and spleen samples from all ASFV-G-ΔI177L-vaccinated pigs were tested for infectious virus using primary porcine macrophage cultures. Infectious virus was detected predominantly in the tonsils or spleen of most animals in each group.

### 2.4. ASFV-G-9GL

A recombinant version of ASFV-G, named ASFV-G-9GLv, was generated by deleting the 9GL (B119L) gene [43]. The intramuscular administration of 10^4^ HAD_50_ of this mutant resulted in a virulent phenotype. However, lower doses (10^2^–10^3^ HAD_50_) did not induce disease and conferred protection against challenge with ASFV-G. Specifically, a dose of 10^2^ HAD_50_ offered partial protection when pigs were challenged at either 21 or 28 dpi, whereas a dose of 10^3^ HAD_50_ provided partial protection at 21 dpi and complete protection by day 28.

The administration of ASFV-G-9GL at 10^2^–10^3^ HAD_50_ did not result in clinical illness. However, pigs inoculated with these doses exhibited variable levels of viremia. At 28 dpi, viremia reached 10^2^ HAD_50_/mL, and, at 21 dpc, it reached 10^7^–10^8^ HAD_50_/mL. The virus was not detected in the nasal cavities of inoculated pigs during the monitoring period. Results regarding ASFV persistence in organs and tissues of surviving animals are not presented.

In some cases, the deletion of a single gene did not lead to the complete attenuation of the virus, especially when attempts were made to increase immunogenicity by using higher viral doses [43].

### 2.5. ASFV-G-ΔA137R

Gladue et al. (2021) demonstrated that the removal of the A137R gene from the highly virulent ASFV-G isolate significantly reduces pathogenicity in pigs [44,45]. Pigs inoculated intramuscularly with 10^2^ HAD_50_ of ASFV-G-ΔA137R remained clinically healthy during the 28-day observation period. However, moderate-to-high levels of viremia were recorded across all vaccinated animals. Viremia levels gradually decreased throughout the experimental period (21 dpc), and the circulating virus was undetectable in the blood of four out of five animals in this group by the end of the experiment. The remaining animal had very low viral titers (10^2.8^ HAD_50_/mL) at 21 dpc. Thus, the loss of virulence in ASFV due to A137R deletion is accompanied by reduced but stable viral replication that manifested as prolonged viremia with relatively low titer values.

The presented results indicate the prolonged replication of attenuated ASFV-G-ΔA137R in pigs. At the end of the experiment, Ct values in tonsils reached 15 and, in the spleen, -22, corresponding to ASFV titers ≥ 10^7^ HAD_50_/0.1 g.

### 2.6. ASFV-GZΔI73R

In vivo evaluations of safety and immunogenicity revealed that the deletion mutant ASFV-GZΔI73R is non-pathogenic and induces robust protection against wild-type ASFV in pigs [46]. In pigs inoculated IM with 10^3^ or 10^5^ TCID_50_ ASFV-GZΔI73R, viremia was very low (10^2.55^–10^3.85^ copies/mL) between 9 and 14 dpi and undetectable between 21 and 28 dpi. Viral shedding was not detected in any of the pigs inoculated with ASFV-GZΔI73R throughout the observation period. Viremia disappeared approximately 4–5 weeks after vaccination. Viral infection did not recur by the end of the 56-day observation period. Furthermore, no viral DNA was detected in any tissue samples, indicating complete clearance of the ASFV-GZΔI73R strain within the 56-day observation period. These results confirm that the ASFV-GZΔI73R strain is sufficiently safe for pig immunization.

At 28 dpi, all ASFV-GZΔI73R-immunized pigs were challenged intramuscularly with 10^4^ TCID_50_ of the parental ASFV-GZ strain. Following challenge, vaccinated pigs developed mild fever (40.0–40.9 °C) lasting 7–8 days along with transient viremia. Post mortem examination revealed generally normal anatomical features, although the slight swelling or hyperemia of certain lymph nodes was noted. Importantly, no viral genetic material was detected in any tissues, confirming the complete elimination of the ASFV-GZΔI73R strain within 28 dpi.

The data presented in Appendix A indicate that mutants derived from ASFV isolates with a single gene deletion generally exhibit high viremia levels following vaccination. Viremia reaching 10^6^–10^7^ HAD_50_/mL by day 28 post vaccination is comparable to the viral loads observed during the early stages of acute and subacute forms of ASF.

## 3. ASF MLV with Two or More Deletions

### 3.1. ASFV-G-MGF

O’Donnell V. et al. (2015) described a recombinant virus generated from the strain ASFV-G by removing six genes belonging to the MGF360 and MGF505 families, resulting in the ASFV-G-MGF mutant [47]. In vivo studies showed that this strain is fully attenuated in pigs: animals inoculated intramuscularly with either 10^2^ or 10^4^ HAD_50_ remained healthy and did not display any clinical signs of disease. Notably, when later challenged with the virulent parental strain, no symptoms of illness were observed, although some pigs became asymptomatic carriers of the challenge virus. Animals vaccinated IM with 10^2^ or 10^4^ HAD_50_ of ASFV-G-MGF were protected against clinical disease after challenge with the virulent strain at 28 dpi. However, viremia levels varied significantly among individuals. At the time of challenge, viral titers in some animals reached 10^3^–10^4^ HAD_50_/mL. All animals receiving 10^2^ or 10^4^ HAD_50_ of ASFV-G-MGF survived infection with the parental virulent virus. Additionally, all these pigs remained clinically normal throughout the observation period (21 dpc), except for transient moderate fever lasting one day in some animals. Viremia was detected (at varying levels) in 80% of animals in both groups after challenge. Despite complete protection against clinical dissemination, approximately 30–40% of ASFV-G-MGF-vaccinated animals retained parental virus infection. Data on ASFV persistence in organs are not provided.

### 3.2. ASFV-ΔH240R-Δ7R

Li J. et al. (2023) developed a live attenuated variant, designated ASFV-ΔH240R-Δ7R, by knocking out the H240R and MGF505-7R genes from the ASFV HLJ/18 isolate [48]. Piglets vaccinated intramuscularly with either 10^3^ or 10^5^ HAD_50_ of this mutant showed no signs of disease and did not transmit the virus through direct contact. Upon challenge with the virulent HLJ/18 strain, piglets vaccinated with 10^3^ HAD_50_ exhibited 100% survival with mild clinical symptoms, while those receiving 10^5^ HAD_50_ remained entirely free of clinical signs. Histopathological and necropsy evaluations confirmed minimal or no tissue damage in the high-dose group. Importantly, only trace amounts of viral genomic material (<10^2^ copies) were found in oral and anal swabs from pigs in the high-dose group, suggesting limited potential for viral shedding after vaccination. This study demonstrates that increasing the dose positively affects the safety and efficacy of ASF MLV. Testing high and ultra-high (≥10^8^) doses of ASF MLV is desirable during candidate vaccine evaluation to ensure their purity and confirm the effectiveness of virus-specific immune mechanisms induced by the end of viremia.

### 3.3. ASFV-MEC-01

Kim M.H. et al. (2024) developed ASFV-MEC-01 by passaging a field isolate in CA-CAS-01-A cells [49]. The resulting strain contains deletions in 12 genes located within the left variable region (LVR). All vaccinated pigs remained afebrile and displayed no clinical symptoms related to ASF during the observation period post vaccination. Following challenge exposure, vaccinated animals experienced a brief febrile response on day 7 post challenge, which resolved shortly afterward. All pigs recovered fully and appeared clinically normal by the end of the 14-day monitoring period. Viremia levels in ASFV-MEC-01-immunized pigs were low-to-moderate in blood, oral, and rectal samples collected by 18 dpi, with further decline observed by 21 dpi. Post mortem examination revealed no evidence of viremia or pathological changes in organs. No macroscopic lesions or clinical abnormalities were identified in pigs vaccinated with ASFV-MEC-01.

### 3.4. ASFV-G-Δ9GL/ΔUK

Studies have demonstrated that the additional deletion of the virulence-associated UK gene significantly enhances the attenuating effect of deleting the 9GL gene. The double-deletion mutant ASFV-G-Δ9GL/ΔUK remained fully attenuated even when administered at doses 100-fold higher than those at which the single-deletion strain ASFV-G Δ9GL exhibited virulent properties (10^4^ HAD_50_) [28]. Regardless of the dose used, pigs inoculated intramuscularly with ASFV-G-Δ9GL/ΔUK did not display any clinical signs associated with ASF and remained healthy throughout the 21-day observation period. A dose of 10^4^ HAD_50_ provided complete protection in all animals following challenge at 28 dpi, while a lower dose of 10^2^ HAD_50_ protected only 40% of the vaccinated pigs. Notably, the administration of a higher dose (10^6^ HAD_50_) appeared less effective compared to 10^4^ HAD_50_: four out of fifteen pigs developed temporary fever, and one animal progressed to severe disease. In contrast, none of the pigs receiving the 10^4^ HAD_50_ dose showed any clinical symptoms after challenge. It is worth noting that elevated viremia levels were observed across all groups both post vaccination and post challenge.

### 3.5. ASFV-G-∆9GL/∆UKp10

Ramirez-Medina E. et al. (2024) demonstrated that ten sequential passages of ASFV-G-∆9GL/∆UK in IPKM (Immortalized Porcine Kidney Macrophage) cells led to minor changes in the viral genome [50]. The protective efficacy of ASFV-G-∆9GL/∆UKp10 was evaluated in pigs inoculated IM with 10^4^ or 10^6^ HAD_50_ of ASFV-G-∆9GL/∆UKp10. Results showed that all groups of animals remained clinically normal throughout the observation period. The pattern of viremia in inoculated animals was heterogeneous in both groups. In some pigs, at 28 dpi, viremia reached 10^3^–10^6^ HAD_50_. While animals inoculated with 10^4^ HAD_50_ were partially protected against experimental infection with the virulent parental ASFV-G, animals inoculated with 10^6^ HAD_50_ were fully protected.

As shown in Tables S1A and S1B, introducing an additional deletion into the ASFV-G-Δ9GL strain did not significantly improve viremia profiles. Double-deletion mutants such as ASFV-G-Δ9GL/ΔUK and ASFV-G-∆9GL/∆UKp10 still exhibited high viremia titers both after vaccination and challenge infection. High viremia levels were also observed among mutants with multiple deletions.

## 4. ASF MLV with Two Deletions, Including EP402R

Of the approximately 70 viral proteins detected in purified ASFV virions, only one—CD2v, encoded by the EP402R gene—has been identified as a component of the outer viral envelope through proteomic studies [51,52]. CD2v plays a key role in the attachment of erythrocytes to infected cells and extracellular virions, a process known as hemadsorption (HAd) [53,54,55]. However, this protein is not essential for infection, and naturally occurring isolates lacking hemadsorption activity have been obtained through the targeted deletion of the EP402R gene. In pigs infected with HAd-positive strains, over 90% of viral particles are found to be associated with the erythrocyte fraction [53], which may help shield both free virus and infected cells from immune recognition, thereby limiting viral clearance [56].

Evidence suggests that CD2v also contributes to cross-protective immunity. Based on the ability of attenuated strains to induce protection against different ASFV genotypes, nine distinct seroimmunotypes have been defined [7]. This correlates with the capacity of sera from vaccinated pigs to inhibit hemadsorption. Since CD2v is essential for inducing HAd, its genetic sequence may serve as a useful predictor of cross-protection among immunotypes. Removing the CD2v gene from highly virulent strains such as Ba71 [57] or Kenya-IX-1033 [58] has been shown to either completely or partially reduce their pathogenic potential, respectively. Several studies have aimed at further reducing the virulence of modified live ASFV vaccines and improving their safety profile by introducing an additional deletion targeting the second virulence-associated gene, EP402R.

### 4.1. ASFV-SY18-∆CD2v/UK

Teklue T. et al. (2020) developed a double-deletion mutant, ASFV-SY18-∆CD2v/UK [59]. As anticipated, this strain lost its ability to form characteristic erythrocyte rosettes around infected macrophages due to the absence of CD2v. More importantly, pigs inoculated with ASFV-SY18-∆CD2v/UK remained clinically healthy throughout a 28-day observation period. No viral genome was detected in blood samples or nasal swabs after vaccination. Following challenge with 10^4^ TCID_50_ of wild-type ASFV-SY18, all previously vaccinated pigs maintained normal body temperature and showed no signs of illness. However, residual ASFV DNA was detected post challenge, with weakly positive PCR signals observed in blood and lymphoid tissues. These findings suggest incomplete clearance of the parental ASFV-SY18 strain in some animals.

### 4.2. SY18∆L60L∆CD2v

The deletion of the L60L gene from the ASFV SY18 genome leads to partial attenuation; however, survival rates were only 60% when pigs were vaccinated with a high dose of SY18∆L60L [60]. In contrast, the administration of a higher dose (10^5^ TCID_50_) of SY18∆L60L∆CD2v resulted in full protection against challenge with 10^2^ TCID_50_ of the virulent SY18 strain at 21 dpc [61]. Lower doses (10^2^ TCID_50_) of SY18∆L60L∆CD2v offered limited protection, with only one out of five pigs surviving challenge with the parental SY18 isolate. None of the vaccinated pigs exhibited clinical signs of ASF during the 28-day monitoring period, and all animals maintained normal body temperature and general health. Viral genome levels in blood were either very low or undetectable, with viremia titers below 1 × 10^2^ copies/mL in SY18∆L60L∆CD2v-vaccinated animals. Results indicate that pigs receiving the high-dose vaccination had minimal viral replication following SY18 challenge. Viral loads in blood, oral, and rectal swabs remained consistently below 10^2^ copies/mL. Compared to pigs infected with the wild-type virus, those vaccinated with SY18∆L60L∆CD2v showed either negligible or undetectable levels of viral genome in circulation. Histopathological examination revealed no abnormalities in liver, spleen, lungs, kidneys, or submandibular lymph nodes of surviving pigs. However, up to 10^3^ copies/mL of the p72 gene were detected in the heart, kidney, and thymus tissue, suggesting the presence of persistent virus. Thus, despite low-level viremia, evidence of viral persistence was still observed.

### 4.3. Arm-∆CD2v-∆A238L

Pérez-Núñez D. et al. (2022) [62] developed a vaccine candidate based on the genotype II strain Arm/07/CBM/c2 by introducing targeted deletions in the EP402R and A238L genes using CRISPR/Cas9 technology in COS-1 cells. No unintended genetic modifications were detected in the resulting mutant [62]. A group of pigs was inoculated IM with 10^2^ TCID_50_ per animal of Arm-∆CD2v-∆A238L. Animals were monitored for 28 days for ASF clinical signs. All pigs showed no significant increase in clinical score throughout the observation period. Viremia levels, Ct ranging from 35.3 to 39.8, were detected in vaccinated pigs on 7–14 dpi. By days 21–28 dpi, ASFV DNA was undetectable in the blood.

After IM challenge with 10^2^ HAD_50_ of the Korean Paju strain, 100% of vaccinated animals remained healthy and alive three weeks post challenge. Accordingly, body temperature in all vaccinated animals remained within normal range. Moreover, clinical scores were consistently below threshold values throughout the 21-day observation period. Low viremia levels were detected between 7 and 21 dpc (end of the experiment).

The virus was absent or present only at very low levels in oral and fecal exudates. Finally, low virus levels were detected in tissues of vaccinated animals (at 21 dpc). Tonsils showed higher virus detection levels compared to other tissues in all vaccinated pigs.

### 4.4. ASFV-G-∆I177L/∆EP402R

The mutant ASFV-G-∆I177L was considered a safe and effective ASF MLV, inducing protection against parental virus infection with the Georgia 2010 isolate [63]. Based on this, a potential DIVA-compatible vaccine candidate was developed by deleting the EP402R gene.

Domestic pigs vaccinated with 10^2^ or 10^6^ HAD_50_ of ASFV-G-∆I177L/∆EP402R remained clinically normal, and infectious virus was not detected in blood samples from vaccinated ASFV-G-∆I177L/∆EP402R animals in any group at any of the tested time points. This may be related to measurements not being conducted between 2 and 6 dpi.

After challenge with 10^2^ HAD_50_ of ASFV-G, all animals vaccinated with ASFV-G-∆I177L/∆EP402R were protected and remained clinically normal until the end of the observation period. However, analyzing viremia kinetics after the challenge of ASFV-G-∆I177L/∆EP402R-vaccinated pigs reveals signs of reduced ASF MLV efficacy. In animals inoculated with ASFV-G-∆I177L, with viremia levels at 28 dpi ranging from 10^3.3^ HAD_50_/mL to 10^6.5^ HAD_50_/mL, viral titers decreased to less than 10^1.8^ HAD_50_/mL by 21 dpc [64]. In contrast, in pigs vaccinated with ASFV-G-∆I177L/∆EP402R, with viremia levels at 28 dpi below 10^1.8^ TCID_50_/mL, viremia increased by day 11 to 10^5^–10^6^ HAD_50_/mL and was in the range of 10^1.8^ to 10^5^–10^7^ HAD_50_/mL by day 21. Data on ASFV persistence in tissues and organs are not presented in this study.

Thus, the deletion of the EP402R gene from ASFV-G-∆I177L led to viremia development after challenge.

### 4.5. BeninΔP148RΔEP402R

In pigs immunized with the BeninΔDP148R virus, the viremia peak was detected in the blood at 5–6 dpi, coinciding with the onset of clinical signs. Clinical signs disappeared afterward, but the viral genome in the blood slowly decreased over approximately 60 days. Infectious virus declined faster and was undetectable by approximately 28–30 days [65]. The deletion of the EP402R gene sharply reduced the persistence of infectious virus in the blood of vaccinated pigs from 28 to 14 days, and the viral genome from 59 to 14 days, while maintaining a high level of protection against infection [66]. Infectious virus was first detected on day 5 dpi and reached a peak level of 10^3.75^–10^4.50^ TCID_50_/mL by day 7. After challenge with the virulent Benin 97/1 isolate, infectious virus was not detected. All animals survived. Aside from mild lymphadenopathy, no other relevant macroscopic lesions were observed in the BeninΔP148RΔEP402R-vaccinated pig group. Unfortunately, the results of ASFV detection in organs and tissues were not provided.

### 4.6. ASFV-G-∆9GL/∆CD2v

Gladue D. P. et al. (2020) [67] investigated enhancing the safety of the experimental vaccine strain ASFV-G-∆9GL by deleting the EP402R and EP153R genes. This led to the development of two new recombinant viruses: ASFV-G-∆9GL/∆CD2v (with two gene deletions) and ASFV-G-∆9GL/∆CD2v/∆EP153R (with three deletions) [67]. Despite inducing minimal or undetectable viremia in vaccinated pigs, both constructs failed to provide protection against challenge with the virulent parental strain ASFV Georgia. In contrast, the original ASFV-G-∆9GL strain offered consistent protection under similar conditions. These findings indicate that removing CD2-like and C-type lectin receptor-binding genes significantly diminishes the immunogenic potential of this MLV candidate.

### 4.7. SY18∆MGF/CD2v

In the pigs vaccinated with SY18∆MGF/CD2v, viremia was not detected after inoculation [38]. Following challenge, viral shedding was frequently detected with high viral DNA copy numbers in almost all SY18DMGF/CD2v-immunized pigs during the early stage. Survival rate for immunized pigs was 60%. As noted in previous studies, prolonged viremia or viral persistence is not uncommon in animals vaccinated with other attenuated ASFV strains [39,40,41], which is believed to be associated with protection against virulent ASFV infection. No pathological changes were observed in surviving pigs immunized with SY18MGF/CD2v.

The qPCR results were positive with high viral copy numbers (10^3.3^–10^8.5^ copies/mL) when testing samples of the lungs, pulmonary ileal lymph node, bone marrow, adrenal glands, and joint fluid of SY18DMGF/CD2v-immunized pigs.

Results from several studies indicate that the deletion of the EP402R gene generally enhances the safety of ASF MLV. However, viremia and ASFV persistence in organs and tissues after challenge suggest insufficient efficacy.

Results from several studies indicate that deletion of the EP402R gene generally enhances the safety profile of ASF MLV (Appendix A). However, detectable viremia and the persistence of viral DNA in organs and tissues after challenge suggest incomplete protection and insufficient efficacy.

Efforts to improve vaccine candidates through the additional deletion of the ∆EP402R gene eliminated clinical signs of ASF following vaccination and led to a significant reduction in both level and duration of viremia (Appendix A). Nevertheless, viremia was still observed following challenge, persisting until the end of the experimental period.

## 5. Potential ASF MLV Candidates

### 5.1. VNUA-ASFV-LAVL2

Truong Q.L. et al. (2023) developed a safe and effective live attenuated vaccine, VNUA-ASFV-LAVL2, through the serial passaging of a field isolate (VNUA ASFV-05L1, genotype II) in primary porcine alveolar macrophages (PAMs, 65 passages) and an immortalized macrophage cell line (3D4/21, 55 passages) [68]. Pigs administered with low (10^2^ HAD_50_) to high (10^5^ HAD_50_) doses of VNUA-ASFV-LAVL2 exhibited only transient mild fever (<40.6 °C) at 7–10 dpi (low dose) and 5–7 dpi (high dose), followed by normal temperature throughout the 28-day observation period. No clinical signs associated with ASF were observed. Peak viremia levels reached 10^4^–10^5^ HAD_50_/mL between days 7 and 11 post vaccination but declined rapidly thereafter, becoming nearly undetectable by day 28. Vaccinated pigs demonstrated high protection levels with 100% survival and health against the virulent virus VNUA-ASFV-05L1. Notably, only very low levels of ASFV were detected in blood up to 5 dpc, and ASFV was not detected in oral fluids or rectal swabs from all vaccinated groups. No pathological signs or lesions were observed in any organs of VNUA-ASFV-LAVL2-vaccinated pigs.

### 5.2. VNUA-ASFV-LAVL3

In a follow-up study, Truong Q.L. et al. (2024) [69] generated another promising vaccine candidate, VNUA-ASFV-LAVL3, by the successive passaging of a virulent genotype II strain (VNUA-ASFV-L2) in an immortalized porcine alveolar macrophage cell line (3D4/21) over 50 passages. The resulting strain, VNUA-ASFV-LAVL3, lost its capacity for hemadsorption [69]. All vaccinated pigs remained asymptomatic and showed no clinical indicators of infection during the 28-day monitoring period. Viral clearance occurred efficiently by 14–17 dpi, even at the highest tested dose (10^5^ TCID_50_). Furthermore, no viral shedding was detected in oral or rectal samples throughout the observation. Vaccination with VNUA-ASFV-LAVL3 induced complete protection against lethal challenge with wild-type genotype II ASFV at all tested doses (10^3^, 10^4^, and 10^5^ TCID_50_).

Importantly, these results demonstrate the effectiveness of both traditional methods of repeated virus passage and recombinant approaches, resulting in optimal safety and efficacy in ASF MLV candidates based on viremia characteristics.

### 5.3. ∆MGF360/505_Stav

Koltsov A. et al. (2024) deleted three genes from the multigene family MGF360 (MGF360-12L, MGF360-13L, and MGF360-14L) and three genes from the multigene family MGF505 (MGF505-2R and partial deletions of MGF505-1R and MGF505-3R) from the genome of the virulent ASFV Stavropol_01/08 strain (genotype II, serotype 8) [70]. Inoculated ∆MGF360/505_Stav pigs did not exhibit any clinical disease symptoms. Low-level viremia (1.3 × 10^2^–5.11 × 10^5^ copies/mL of ASFV genome) was observed shortly after inoculation (3–7 dpi), while viral DNA was undetectable at later time points. Furthermore, the ASFV genome was not detected in the organs of inoculated animals that were humanely euthanized at 24 dpi; 100% of ∆MGF360/505_Stav-vaccinated animals were fully protected against homologous ASFV Stavropol_01/08 (10^3^ HAD_50_) infection. The authors did not detect viral genomes of either the virulent or recombinant ASFV strains in blood or organs after challenge.

### 5.4. ASFV-G-∆MGF

Deutschmann P. et al. (2022) [71] evaluated the efficacy of the vaccine candidate ASFV-G-∆MGF. A total of five pigs were vaccinated intramuscularly (IM) twice with a 21-day interval using either 10^4^ (Group 1) or 10^3^ HAD_50_ (Group 2) of ASFV-G-∆MGF. The virus used in Group 1 was propagated in primary macrophages, while the virus in Group 2 was grown on a commercial immortalized cell line. All animals were challenged with the virulent ASFV-Armenia 2008 strain at a dose of 10^4^ HAD_50_ [71].

No clinical signs were observed after vaccination or challenge. In Group 1, only two out of five pigs showed minimal amounts of ASFV DNA: one animal at 7 dpi and the second at 21 dpi. After challenge, viral DNA was detected in two other animals: one at 4 dpc and another at 10 and 14 dpc. In Group 2, low levels of viral DNA were also detected in two pigs: one at 7 dpi and the other at 14 dpi. All remaining samples tested negative by PCR. At the end of the study, all animals survived, and only one pig from Group 1 had minimal traces of ASFV DNA detected in organ tissues.

The findings summarized in Appendix A demonstrate the feasibility of developing modified live ASF vaccines that not only prevent clinical signs of disease and provide 100% protection against virulent challenge, but also show no detectable viremia by day 28 post vaccination or post challenge.

## 6. Discussion

Safe and effective ASF MLVs can be obtained either through the serial passaging of the virus in primary and/or immortalized cell cultures or by deleting genes associated with virulence. The identification and controlled modification of key virulence determinants has now become a standard strategy in the development of experimental, live, and attenuated vaccine candidates. ASF MLV obtained via the deletion of a single gene may be safe in terms of clinical signs and effective in terms of survival, but most exhibit viremia at 28 dpi and 21 dpc. Clinical signs of disease often reappear after challenge with a virulent strain. Post mortem examination of organs indicates ASFV persistence [38,44].

Safety and efficacy studies of most such ASF MLVs across a wide range of doses have not yielded significant positive results. It is important to note that ASF MLV should be tested across a broad dose range from 10^2^ to 10^7^ HAD_50_ (TCID_50_) or higher. This would confirm viral purity and provide insights into the timing of immune protection formation. Typically, increasing dosage shortens the time required for immune protection development [38,45]. Studies of ASF MLV with a single-gene deletion are likely more relevant for determining the functional role of the gene product in virulence or other viral properties.

Efforts have been made to obtain ASF MLV through the deletion of two or more genes. Vaccinated pigs lacked clinical signs both after vaccination and challenge, with survival rates reaching up to 100%. However, viremia was noted in most studies both after vaccination and challenge. Infectious ASFV was detected in the organs of surviving animals upon euthanasia in most cases [28,48,49]. In particular, the new strain ASFV-G-∆I177L/∆LVR, derived from the production strain ASFV-G-ΔI177L through passage in an immortalized cell culture, proved to be equivalent in terms of safety, immunogenicity, and protective efficacy. However, viremia persisted in all pigs vaccinated with ASFV-G-∆I177L/∆LVR up to day 28, with relatively high titers observed in some cases. At 21 days post challenge with the virulent ASFV-G isolate, individual animals in groups vaccinated with either 10^2^ or 10^4^ HAD_50_ of ASFV-G-∆I177L/∆LVR showed viremia levels reaching 10^4^–10^5^ HAD_50_/mL [72]. The absence of detailed data on viral persistence in organs of pigs vaccinated with ASFV-G-∆I177L/∆LVR and who were subsequently challenged limits the ability to objectively assess the strain’s potential for further vaccine development.

The additional deletion of the EP402R gene (CD2v) enhances the safety of ASF MLV due to rapid clearance. Some studies reported reduced clinical signs of ASF [66]. In most works, viremia was not detected after vaccination, while in some, it persisted for no longer than 7–14 dpi [61,62,63]. Since the deletion of the EP402R gene does not significantly affect ASFV replication in vitro, reduced viremia is likely due to decreased viral spread caused by virion binding to red blood cells. Notably, Borca et al. [39,63] published intriguing results. After inoculation with ASFV-Georgia-ΔI177L, viremia reached 10^5^–10^6^ HAD_50_ at 28 dpi, whereas, after inoculation with ASFV-G-∆I177L/∆EP402R, viremia was completely absent from day 0 to 28 dpi. Conversely, after challenge with a virulent ASFV-G strain, viremia was absent in the ASFV-Georgia-ΔI177L group at 21 dpc [39]; but, in the ASFV-G-∆I177L/∆EP402R group, viremia in some animals reached 10^5^–10^6^ HAD_50_/mL [63]. No differences were observed in clinical signs or survival rates. While animal survival after challenge indicated protective immunity formation, viremia and ASFV persistence in organs and tissues were noted in ASFV-G-∆I177L/∆EP402R groups after challenge. This correlation was also evident when the immunobiological features of the Lv17/WB/Rie1-∆CD mutant were compared to those of the wild-type Lv17/WB/Rie1 strain [73]. Therefore, obtaining promising ASF MLV with a second deletion in the EP402R gene as a candidate vaccine remains problematic.

Why? CD2v is the major membrane protein and the only virus-specific protein located on the outer envelope of ASFV [19,20,74]. It is also the only one among the 70 viral proteins with serotype-specific characteristics [75]. This suggests that CD2v may belong to the pool of protective proteins. Since protection induced by ASF MLV is seroimmunotype-specific [75], an attenuated mutant with an additional deletion in the EP402R gene induces an immune response sufficient to protect animals from disease and death but insufficient to eliminate viremia and the viral persistence of the challenging virulent isolate/strain.

Regarding viremia duration, there is an opinion that high-level viremia for prolonged periods after ASF MLV administration in pigs (≥28 days) is common and not critical [38,40,41,76]. We believe this position requires further discussion. First, the peak of key effector mechanisms of virus-specific immune protection—antibody-dependent cellular cytotoxicity (ADCC) and cytotoxic T lymphocytes (CTL)—occurs between 3 and 15 dpi [77,78,79,80,81,82]. This implies that ASF MLV replication should be substantially restricted between 14 and 28–35 dpi. Second, viremia and associated viral shedding and persistence in tissues and organs increase the risks of the horizontal and vertical transmission of ASF MLV, as well as recombination with other attenuated or virulent ASFV strains/isolates [83,84,85,86,87]. Thus, viremia 21–28 days after vaccination or challenge is an extremely unfavorable indicator for candidate ASF vaccines.

This concern is supported by data from studies involving the attenuated FK-32/135 strain, derived via the serial passaging of the virulent France-32 isolate in primary cultures of pig bone marrow cells. Vaccination with FK-32/135 induced protection against virulent ASFV strains of seroimmunotype IV. On day 7 after the IM administration of the FK-32/135 strain at a dose of 10^7^ HAD_50_, the virus (10^1^–10^2^ HAD_50_/mL) was detected in pig blood and, by day 14, the virus was undetectable. Rapid protection was achieved by administering a highly concentrated (100×) vaccine formulation at 10^9^ HAD_50_ via the intramuscular route. Pigs were intramuscularly challenged with the virulent France-32 strain at a dose of 10^4^ HAD_50_ at 1, 3, 5, or 7 dpi. Protection was established in 100% of pigs after three, five, and seven days after vaccination [85]. The France-32 strain causes pig death at 6 dpc without detectable CTL or ADCC responses. In pigs inoculated with the FK-32/135 strain at a dose of 10^8^ HAD_50_, CTL-mediated cytolysis reached 26% at 6 dpi, and ADCC ranged from 26% to 28% at 3–8 dpi. Viremia was 10^0.5^ HAD_50_/mL at 8 dpi [80].

There are reports of ASF MLV with favorable safety, efficacy, and viremia levels and durations. These have been obtained both through targeted deletions and passaging in cell cultures [68,69,70]. Whole-genome sequencing and analysis at passage 120 revealed that VNUA-ASFV-LAVL2 has deletions of 13 known genes and 14 uncharacterized deletions in the MGF region of VNUA-ASFV-LAVL2 [68]. In the modified strain ∆MGF360/505_Stav, three genes from the multigene family MGF360 (MGF360-12L, MGF360-13L, and MGF360-14L) and three genes from the multigene family MGF505 (MGF505-2R and partial deletions of MGF505-1R and MGF505-3R) were deleted from the genome of the virulent ASFV Stavropol_01/08 strain [70]. To date, over 130 attenuated ASFV strains have been studied [88]. Only a few have been evaluated against most criteria of purity, activity, biosafety, and efficacy. Deletions in MGF regions allow for the development of safe and effective ASF MLV with limited viremia levels and durations. Combining these deletions with point mutations in the CD2v gene may lead to promising ASF MLV candidates.

It must be acknowledged that the absence of clinical signs and the survival of vaccinated pigs alone do not determine the prospects of ASF MLVs. Here, we propose viremia testing as an essential criterion for evaluating candidate ASF vaccines, with levels exceeding 10^4^ HAD_50_/TCID_50_ and lasting longer than 21–28 days post vaccination considered unfavorable indicators.

However, long-term biosafety under field conditions remains under careful evaluation. Concerns are growing among specialists, administrators, and farmers about potential further mutations and recombination events between vaccine strains and epidemic viruses.

## 7. Conclusions

In conclusion, this work advocates for stricter evaluation criteria for candidate ASF vaccines. We demonstrate that the absence of clinical signs and the survival of vaccinated pigs before and after challenge are insufficient for the preliminary determination of MLV potential. Instead, we propose that the level and duration of viremia after both vaccination and challenge should be considered key parameters at early stages of MLV candidate selection. Importantly, applying these criteria allows for the narrowing of the search for promising MLV candidates. These findings are particularly relevant in the context of accelerating vaccine development efforts that aim for a rapid response in ASF-affected areas.

## Data Availability

The original contributions presented in this study are included in the article/Appendix A. Further inquiries can be directed to the corresponding authors.

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
