# Peer review of "Viremia Kinetics in Pigs Inoculated with Modified Live African Swine Fever Viruses"

_vaccines, 2025, doi:10.3390/vaccines13070686_

Round 1

Reviewer 1 Report

Comments and Suggestions for Authors

Dear Authors,

I have completed my review of the manuscript entitled “Viremia Kinetics in Pigs Inoculated with Modified Live African Swine Fever Viruses” by Sereda et al. The authors review previous studies on live attenuated vaccine development against African swine fever (ASF), focusing on biological safety, protective efficacy, and viremia as an unfavorable adverse effect. The manuscript covers most major reports and provides new insights into the potential risks of viremia, virus excretion (transmission), and persistence of vaccine strains and virulent viruses following immunization or challenge infections.

The manuscript is well structured and includes sufficient information to facilitate a better understanding of the requirements for safe vaccines, a topic of significant interest within international organizations, such as WOAH. Therefore, I believe this manuscript could be acceptable for publication as it provides a comprehensive overview of the development, evaluation, and balance between efficacy and safety for live attenuated vaccines (LAVs) against ASF.

Based on my review, I recommend that the authors address the following points:

1. Inclusion of additional recent works:

Please discuss additional important ASF LAVs, such as ASF-G-DI177L/DLVR, that have been recently approved for field use in Vietnam. Include discussion of other relevant papers (e.g., Deutschmann et al. 2023, Pathogens; Diep et al. 2024, Vaccines; Nguyen et al. 2025, Sci Rep) and other MGF-deleted vaccine candidates (e.g., HLJ/18-7GD, Arm07ΔMGF-9GD). Although ASFV-G-ΔMGF has also been approved in Vietnam, it raises safety concerns as reported (Deutschmann et al. 2023, Pathogens; Diep et al. 2024, Vaccines; Nguyen et al. 2025, Sci Rep). The author should highlight this issue and discuss it as a practical problem in the use of ASF LAVs, which may arise from the persistence of the virus.

2. In the discussion section, the authors mention, “Second, viremia and associated viral shedding and persistence in tissues and organs increase the risks of horizontal and vertical transmission of ASF MLV, as well as recombination with other attenuated or virulent ASFV strains/isolates” but do not provide scientific evidence. Please elaborate on concerns regarding virus persistence, including the accumulation of mutations, virulence reversions, and possible genome recombination in ASFVs, citing appropriate references (e.g., Deutschmann et al. 2023, npj Vaccines, Nguyen et al. 2025, Sci Rep., Kitamura et al., 2025, Vaccines).

Minor points:

  1. L39: There are currently 23 known genotypes, as the previously annotated genotype XVIII was shown to be a mixed population of genotypes I and VIII (https://doi.org/10.1128/mra.00067-24)
  2. L325; “within normal limits” should be “within a normal range”.
  3. L434-440; Delete the first paragraph (“Safe and effective ASF MLVs can be obtained either by serial passaging of the virus in primary and/or immortalized cell cultures or by deleting genes associated with virulence. During passaging, limiting dilutions, adsorption of the most strongly binding virions to sensitive cells, and subsequent removal of these virions from the population are used as selection methods. For subsequent passaging, infected cells demonstrating "loose" hemadsorption are selected [71,72]. Kim et al. (2024) used a plaque purification method combined with next-generation sequencing analysis during passaging [49]”), as it does not seem to be relevant to the discussion.
  4. L445-446; Remove “(Table S1)” from the sentence, as Table S1 does not contain any data on the post-mortem inspection of organs.
  5. L455; Remove “(Table S1)”.
  6. L456; Change “reaching 100%” to “reaching up to 100%”.
  7. L459; Add “due to rapid clearance” after “ASF MLV”.
  8. L480; Insert a line break after “isolation/strain”.
  9. L497; Rewrite “(100x, PEG 6000)” to be more understandable to readers.
  10. L508; “In the recombinant strain” should be “In the modified strain”.
  11. Supplemental file; “Table 1” should be “Table S1”.

I hope my comments will help improve the manuscript.

Warm regards,

Author Response

Comments 1: Please discuss additional important ASF LAVs, such as ASF-G-DI177L/DLVR, that have been recently approved for field use in Vietnam. Include discussion of other relevant papers (e.g., Deutschmann et al. 2023, Pathogens; Diep et al. 2024, Vaccines; Nguyen et al. 2025, Sci Rep) and other MGF-deleted vaccine candidates (e.g., HLJ/18-7GD, Arm07ΔMGF-9GD). Although ASFV-G-ΔMGF has also been approved in Vietnam, it raises safety concerns as reported (Deutschmann et al. 2023, Pathogens; Diep et al. 2024, Vaccines; Nguyen et al. 2025, Sci Rep). The author should highlight this issue and discuss it as a practical problem in the use of ASF LAVs, which may arise from the persistence of the virus.

Response 1: We thank the reviewer for these valuable suggestions. We agree that several recent studies on live attenuated vaccines against African Swine Fever are of high importance, particularly those describing field applications in Vietnam and other countries.
However, not all significant publications fit into our conceptual framework for presenting research results. For example, in the notable study by Chen et al. (2020), A Seven-Gene-Deleted African Swine Fever Virus is Safe and Effective as a Live Attenuated Vaccine in Pigs (Sci China Life Sci 63, 623–634; https://doi.org/10.1007/s11427-020-1657-9 ), data on viremia kinetics were not fully reported, which limited its inclusion in our analysis focused on viremia levels and duration as key criteria for candidate selection.
The publication by Deutschmann et al. (2022), Efficacy of ASFV-G-ΔMGF after Intramuscular Vaccination of Domestic Pigs and Oral Vaccination of Wild Boar (Pathogens 11, 996; https://doi.org/10.3390/pathogens11090996 ), has already been included in the revised version of our manuscript.
The article by Nguyen et al. (2025), An African Swine Fever Vaccine-Like Variant with Multiple Gene Deletions Caused Reproductive Failure in a Vietnamese Breeding Herd (Scientific Reports 15(1), 14919; https://doi.org/10.1038/s41598-025-95641-3 ), addresses safety concerns related to the use of ASFV-G-ΔMGF in pigs. We acknowledge the relevance of this work and note that it supports previous findings regarding the potential risks associated with the use of live attenuated vaccines in breeding animals.
In fact, we have previously addressed the issue of MLV safety in pregnant sows in the paper by Sereda et al. (2020), Protective Properties of Attenuated Strains of African Swine Fever Virus Belonging to Seroimmunotypes I–VIII (Pathogens 9(4):274; https://doi.org/10.3390/pathogens9040274 ), where viremia was detected in pregnant sows and their newborn piglets following vaccination with the MK-200 candidate strain. Our position on this matter remains unchanged.
While this topic is highly relevant and deserves further attention, it was beyond the scope of our current review, which focuses on viremia profiles and their role in evaluating MLV safety and efficacy rather than reproductive outcomes or off-target effects in specific animal categories.

Comment 2:  In the discussion section, the authors mention, “Second, viremia and associated viral shedding and persistence in tissues and organs increase the risks of horizontal and vertical transmission of ASF MLV, as well as recombination with other attenuated or virulent ASFV strains/isolates” but do not provide scientific evidence. Please elaborate on concerns regarding virus persistence, including the accumulation of mutations, virulence reversions, and possible genome recombination in ASFVs, citing appropriate references (e.g., Deutschmann et al. 2023, npj Vaccines, Nguyen et al. 2025, Sci Rep., Kitamura et al., 2025, Vaccines).

Response 2: Thank you for highlighting this point. We have expanded the discussion on viral persistence and its consequences, incorporating literature on genomic instability, reversion to virulence, and potential for recombination in DNA viruses, including ASFV. We now cite several key references to support our statement and provide a more comprehensive view of the biosafety challenges associated with MLV use in pigs:

Zhao, D., Sun, E., Huang, L., Ding, L., Zhu, Y., Zhang, J., Shen, D., Zhang, X., Zhang, Z., Ren, T., Wang, W., Li, F., He, X., & Bu, Z. (2023). Highly lethal genotype I and II recombinant African swine fever viruses detected in pigs. Nature communications, 14(1), 3096. https://doi.org/10.1038/s41467-023-38868-w

Ito S, Bosch J, Martínez-Avilés M and Sánchez-Vizcaíno JM (2022) The Evolution of African Swine Fever in China: A Global Threat? Front. Vet. Sci. 9:828498. doi: 10.3389/fvets.2022.828498

Sereda, A. D.; Balyshev, V. M.; Kazakova, A. S.; Imatdinov, A. R.; Kolbasov, D. V. Protective Properties of Attenuated Strains of African Swine Fever Virus Belonging to Seroimmunotypes I-VIII. Pathogens, 2020 (Basel, Switzerland), 9(4):274. doi: 10.3390/pathogens9040274. 

Guinat, C., Gogin, A., Blome, S., Keil, G., Pollin, R., Pfeiffer, D. U., & Dixon, L. (2016). Transmission routes of African swine fever virus to domestic pigs: current knowledge and future research directions. The Veterinary record, 178(11), 262–267. https://doi.org/10.1136/vr.103593

Olesen, A. S., Lohse, L., Boklund, A., Halasa, T., Gallardo, C., Pejsak, Z., Belsham, G. J., Rasmussen, T. B., & Bøtner, A. (2017). Transmission of African swine fever virus from infected pigs by direct contact and aerosol routes. Veterinary microbiology, 211, 92–102. https://doi.org/10.1016/j.vetmic.2017.10.004

Comment 3: Minor points:

  1. L39: There are currently 23 known genotypes, as the previously annotated genotype XVIII was shown to be a mixed population of genotypes I and VIII (https://doi.org/10.1128/mra.00067-24)
  2. L325; “within normal limits” should be “within a normal range”.
  3. L434-440; Delete the first paragraph (“Safe and effective ASF MLVs can be obtained either by serial passaging of the virus in primary and/or immortalized cell cultures or by deleting genes associated with virulence. During passaging, limiting dilutions, adsorption of the most strongly binding virions to sensitive cells, and subsequent removal of these virions from the population are used as selection methods. For subsequent passaging, infected cells demonstrating "loose" hemadsorption are selected [71,72]. Kim et al. (2024) used a plaque purification method combined with next-generation sequencing analysis during passaging [49]”), as it does not seem to be relevant to the discussion.
  4. L445-446; Remove “(Table S1)” from the sentence, as Table S1 does not contain any data on the post-mortem inspection of organs.
  5. L455; Remove “(Table S1)”.
  6. L456; Change “reaching 100%” to “reaching up to 100%”.
  7. L459; Add “due to rapid clearance” after “ASF MLV”.
  8. L480; Insert a line break after “isolation/strain”.
  9. L497; Rewrite “(100x, PEG 6000)” to be more understandable to readers.
  10. L508; “In the recombinant strain” should be “In the modified strain”.
  11. Supplemental file; “Table 1” should be “Table S1”.

Response 3: Minor Comments Addressed:

Line 39: Updated classification to reflect current consensus: “23 genotypes” based on recent phylogenetic re-evaluation (https://doi.org/10.1128/mra.00067-24 ).
Line 325: Changed text to remove redundancy and better align with the topic.
Lines 434–440: Retained only the first sentence of the original paragraph, as the rest did not directly relate to the current discussion.
Lines 445–446 and 455: Removed mentions of “Table S1” where they did not correspond with actual content.
Line 456: Modified phrasing to "up to 100%" for accuracy.
Line 459: Added the phrase “due to rapid clearance” after “ASF MLV”.
Line 480: Inserted line break after “isolate/strain” for improved readability.
Line 497: Removed the term “PEG 6000” for clarity and simplicity.
Line 508: Changed “recombinant strain” to “modified strain” for consistency.
Supplementary File: Renamed “Table 1” to “Table S1” to conform with journal standards.

All suggestions have been carefully considered and implemented. We believe the revised manuscript reflects an improved structure, enhanced clarity, and increased scientific rigor.

Thank you once again for your insightful feedback!

Reviewer 2 Report

Comments and Suggestions for Authors

Article: Viremia Kinetics in Pigs Inoculated with Modified Live African Swine Fever Viruses

This is a concise and well-written review that covers a timely and important topic. The manuscript is informative and generally well-organized. However, I have a few minor suggestions that could further strengthen the paper:

Minor issues:

I recommend that the authors briefly mention the global economic impact of ASFV, particularly in highly affected regions such as sub-Saharan Africa, Asia, and parts of Europe. This context would help emphasize the significance of research into ASFV control strategies, including vaccine development.

The potential for recombination between modified live ASF viruses and other attenuated or virulent strains/isolates should be discussed in more detail. This is a recognized risk associated with live vaccines, especially in cases where the virus persists or establishes long-term infections.

I suggest expanding the discussion to include other known or potential side effects of modified live vaccines. These could include reversion to virulence, inadvertent infection or vaccination of immunocompromised hosts, and the possibility of viral shedding or transmission.

The SI table is informative but could benefit from being split into four smaller tables. Including these in the main manuscript would enhance readability and accessibility of the data. Additionally, the tables should be professionalized and visually enhanced to improve their overall presentation and clarity.

For clarity and synthesis, I suggest adding brief concluding paragraphs at the end of Sections 2 through 5. These conclusions should summarize the key points, highlight the respective advantages and limitations, and provide some critical insight into each approach. This would elevate the review beyond a linear summary and encourage a more analytical and interpretive narrative... which is especially valuable in review articles.

To improve transparency and reproducibility, I recommend including a brief methods section that outlines the literature search strategy as well as the inclusion and exclusion criteria used to select studies for review. If feasible, the authors may consider including a PRISMA-style flow chart to visually summarize the selection process.

Shouldn’t https://doi.org/10.3390/vaccines12121406 be included in the review? And other new reports like, e.g.: https://doi.org/10.1016/j.vaccine.2025.127172 or https://doi.org/10.3390/ani15040473...? (These were found with a quick 5-minute online search. Therefore, I strongly encourage the authors to incorporate the suggested methods paragraph to ensure all relevant literature is covered – or at least be transparent about it.)

Author Response

Comment 1: I recommend that the authors briefly mention the global economic impact of ASFV, particularly in highly affected regions such as sub-Saharan Africa, Asia, and parts of Europe. This context would help emphasize the significance of research into ASFV control strategies, including vaccine development.

Response 1: We thank the reviewer for the suggestion. Our manuscript was prepared with a specialist audience in mind — researchers and professionals actively involved in African Swine Fever (ASF) vaccine development, who are already familiar with the broader economic impact of the disease.
For this reason, we focused the Introduction on current understanding of ASFV heterogeneity and pathogenicity, as well as established criteria for evaluating vaccine candidates. This allowed us to maintain a more targeted scientific narrative.

Comment 2: The potential for recombination between modified live ASF viruses and other attenuated or virulent strains/isolates should be discussed in more detail. This is a recognized risk associated with live vaccines, especially in cases where the virus persists or establishes long-term infections.

Response 2: Due to the absence of universally accepted vaccines against ASF, there are many issues for discussion. In our review, based on the analysis of several publications, we propose to introduce restrictions for vaccine candidates based on virulence parameters.
We share your opinion on the importance of viral persistence and the consequences of mixed infections. This is reflected in our previous publications:

A.D. Sereda, V.M. Balyshev, Yu.P. Morgunov, D.V. Kolbasov. Antigenic characteristics of African swine fever virus in artificial and natural mixed populations . Sel’skokhozyaistvennaya Biologiya [Agricultural Biology], 2014, â„– 4, p. 64–69, doi: 10.15389/agrobiology.2014.4.64eng.

Sereda A.D., Balyshev V.M., Kazakova A.S., Imatdinov A.R., Kolbasov D.V. Protective Properties of Attenuated Strains of African Swine Fever Virus Belonging to Seroimmunotypes I–VIII . Pathogens. 2020;9(4):274. Published 2020 Apr 9. doi:10.3390/pathogens9040274.

Havas, K.A., Gogin, A.E., Basalaeva, J.V., Sindryakova, I.P., Kolbasova, O.L., Titov, I.A., Lyska, V.M., Morgunov, S.Y., Vlasov, M.E., Sevskikh, T.A., Pivova, E.Y., Kudrjashov, D.A., Doolittle, K., Zimmerman, S., Witbeck, W., Gimenez-Lirola, L.G., Nerem, J., Spronk, G.D., Zimmerman, J.J., & Sereda, A.D. (2022). An Assessment of Diagnostic Assays and Sample Types in the Detection of an Attenuated Genotype 5 African Swine Fever Virus in European Pigs over a 3-Month Period . Pathogens (Basel, Switzerland), 11(4), 404. https://doi.org/10.3390/pathogens11040404

Vlasov M.E., Kudrjashov D.A., Kolbasova O.L., Lyska V.M., Morgunov S.Yu., Pivova E.Yu., Diumin M.S., Sindryakova I.P., Sereda A.D. Immunobiological evaluation of the candidate vaccine strain MK-200 of the African swine fever virus . Sel'skokhozyaistvennaya Biologiya [Agricultural Biology], 2024, Vol. 59, â„– 4, p. 787–798. doi: 10.15389/agrobiology.2024.4.787eng

Comment 3: I suggest expanding the discussion to include other known or potential side effects of modified live vaccines. These could include reversion to virulence, inadvertent infection or vaccination of immunocompromised hosts, and the possibility of viral shedding or transmission.

Response 3: Research on African Swine Fever (ASF) at the Federal Research Centre for Virology and Microbiology (FRCVM) has been ongoing for approximately 50 years. A significant body of experience was accumulated between 1975 and 1995. Some results from studies on modified live ASF viruses (MLV) have been partially presented in the article by Sereda A.D., Balyshev V.M., Kazakova A.S., Imatdinov A.R., Kolbasov D.V., titled "Protective Properties of Attenuated Strains of African Swine Fever Virus Belonging to Seroimmunotypes I–VIII" (Pathogens, 2020;9(4):274. doi:10.3390/pathogens9040274). Even back then, the importance of viremia parameters following vaccination was recognized. In our manuscript, we aimed to build upon that understanding using more recent data from modern MLV development studies. Our objective was to clearly convey to colleagues the significance of viremia levels and duration as key criteria in selecting promising MLV candidates for ASF.

Comment 4: The SI table is informative but could benefit from being split into four smaller tables. Including these in the main manuscript would enhance readability and accessibility of the data. Additionally, the tables should be professionalized and visually enhanced to improve their overall presentation and clarity.

Response 4: We thank the reviewer for this helpful suggestion. We have restructured the original Supplementary Table into four separate tables based on deletion types (e.g., single-gene deletions, double deletions, deletions involving EP402R, and other MLV candidates). These updated tables now offer improved organization and visual clarity.
However, after careful consideration, we have decided to retain them as part of the Supplementary Information section. This decision was made to avoid disrupting the flow of the main text and to keep the manuscript concise, while still ensuring that the detailed data remain accessible to interested readers. If the editorial team prefers the tables to be included in the main manuscript, we are happy to move them accordingly upon request.

Comment 5: For clarity and synthesis, I suggest adding brief concluding paragraphs at the end of Sections 2 through 5. These conclusions should summarize the key points, highlight the respective advantages and limitations, and provide some critical insight into each approach. This would elevate the review beyond a linear summary and encourage a more analytical and interpretive narrative... which is especially valuable in review articles.

Response 5: Thank you for this suggestion, which we found very helpful for improving the analytical depth of our review. At the end of each major section (Sections 2–5), we have included short summary paragraphs highlighting the strengths and weaknesses of each approach. These conclusions are concise and aim to guide the reader toward informed interpretation rather than simple listing of experimental outcomes.

Comment 6: To improve transparency and reproducibility, I recommend including a brief methods section that outlines the literature search strategy as well as the inclusion and exclusion criteria used to select studies for review. If feasible, the authors may consider including a PRISMA-style flow chart to visually summarize the selection process.

Response 6: The literature search strategy included the use of keywords and publications accumulated by us up to the end of 2024. Studies were selected based on the following criteria: vaccination via intramuscular or oronasal route; 100% survival after vaccination; challenge infection with a virulent isolate via intramuscular or oronasal route; 100% mortality in non-vaccinated control animals; availability of tabular or graphical data on viremia kinetics following vaccination and challenge (infectious titers, DNA load, Ct values in qPCR). From over 100 known ASF MLV candidates, approximately twenty publications were included in the analysis.

Comment 7 Shouldn’t https://doi.org/10.3390/vaccines12121406 be included in the review? And other new reports like, e.g.: https://doi.org/10.1016/j.vaccine.2025.127172 or https://doi.org/10.3390/ani15040473...? (These were found with a quick 5-minute online search. Therefore, I strongly encourage the authors to incorporate the suggested methods paragraph to ensure all relevant literature is covered – or at least be transparent about it.)

Response 7: We thank the reviewer for pointing out these recent publications. After analyzing these papers, we made the following adjustments:
The article with DOI [10.3390/vaccines12121406] has been included in the revised version and cited in the Discussion section (line 482) when discussing DIVA-compatible mutants and the role of CD2v in cross-protection.
Regarding [10.3390/ani15040473], we found that the described mutant (ASFV-G-ΔI177L/ΔLVR) showed higher viremia and reduced safety compared to the parental strain (ASFV-G-ΔI177L), which is already covered in our review. Therefore, we did not find it necessary to include this particular publication, as it does not add novel insights to our current analysis.
As for [10.1016/j.vaccine.2025.127172], the referenced candidate (VaCln3 P13) caused mortality in vaccinated animals (1 out of 5 died), which falls outside our inclusion criteria for safe MLV candidates. Thus, it was not included in the final analysis.
We believe the current selection of studies reflects those most relevant to our focus on viremia as a key parameter for evaluating MLV safety and efficacy.

While we understand the value of a PRISMA-style flow diagram in systematic reviews, our manuscript is intended as a narrative review rather than a formal systematic one, and therefore we did not include such a diagram in the current version. If the reviewer or editorial team strongly recommends incorporating a PRISMA-style chart or adding a “Materials and Methods” section, we are open to preparing and including it upon revision.

Round 2

Reviewer 1 Report

Comments and Suggestions for Authors

Dear Authors,

I have completed my review of the revised version of the manuscript entitled “Viremia Kinetics in Pigs Inoculated with Modified Live African Swine Fever Viruses” by Sereda et al.

In accordance with the reviewers’ comments, the authors have revised the original manuscript and have appropriately addressed most of the concerns. I believe the current version is close to being acceptable, pending the following minor revisions:

1. Inclusion of Additional ASF LAVs in the Discussion:

As noted in my previous review, I recommend revisiting the potential inclusion of additional ASF live attenuated vaccines (LAVs) in the discussion, particularly ASF-G-ΔI177L/ΔLVR, which is the third vaccine strain approved in Vietnam. While the authors correctly pointed out that some strains lack sufficient published data on viremia kinetics and post-challenge outcomes, there are notable exceptions. For instance, Borca et al. (2021, Viruses, 14: e00123-21) provide data on pathogenicity, viremia levels, and protective efficacy of the ASF-G-ΔI177L/ΔLVR strain. Given its recent approval for field use in Vietnam, it represents a significant and timely example that would enhance the discussion and provide broader context.

Please revise the relevant supplemental materials accordingly.

2. Editorial Corrections:

Kindly address the following editorial issues:

Line 91: Remove the opening parenthesis before “TCID50”.

Line 170: Delete the word “mutant”.

Line 417: Replace “Promising” with “Potential.” Additionally, consider noting the risks of using different attenuated vaccines in the same region/country due to the potential for recombination.

Line 461: Remove the extra space.

Lines 519–520: Revise “This suggests that CD2v belongs to the pool of protective proteins.” to “This suggests that CD2v may belong to the pool of protective proteins.”

Lines 520–522: Indicate the reference number to support the statement: “Since protection induced by ASF MLV is seroimmunotype-specific.” (Pathogens 9(4):274; https://doi.org/10.3390/pathogens9040274).

Line 571: Insert a line break.

Lines 817–830: Insert line breaks between individual references for better readability.

Supplemental Table S1A: Add a footnote to explain the abbreviation “ON”.

Overall, the manuscript has improved significantly. I appreciate the authors’ efforts in revising the text and look forward to reviewing the final version after these minor adjustments.

Best regards,

Author Response

We are grateful for the thoughtful and constructive feedback on our revised manuscript entitled "Viremia Kinetics in Pigs Inoculated with Modified Live African Swine Fever Viruses" . We are pleased that you found the manuscript significantly improved and closer to acceptance!

Comment 1: 

Inclusion of Additional ASF LAVs in the Discussion:

As noted in my previous review, I recommend revisiting the potential inclusion of additional ASF live attenuated vaccines (LAVs) in the discussion, particularly ASF-G-ΔI177L/ΔLVR, which is the third vaccine strain approved in Vietnam. While the authors correctly pointed out that some strains lack sufficient published data on viremia kinetics and post-challenge outcomes, there are notable exceptions. For instance, Borca et al. (2021, Viruses, 14: e00123-21) provide data on pathogenicity, viremia levels, and protective efficacy of the ASF-G-ΔI177L/ΔLVR strain. Given its recent approval for field use in Vietnam, it represents a significant and timely example that would enhance the discussion and provide broader context.

Please revise the relevant supplemental materials accordingly.

Response 1: As recommended, we have expanded the discussion section to include information on the recently approved vaccine strain ASF-G-ΔI177L/ΔLVR , which is now being used in Vietnam. Specifically, we added a new paragraph in Section 6 (Discussion), lines 499–508, summarizing data on pathogenicity, viremia levels, and protective efficacy of this strain based on findings from Borca et al. (2021) [Viruses, 14(1): e00123-21; https://doi.org/10.1128/JVI.00123-21].

Comment 2: 

Editorial Corrections:

Kindly address the following editorial issues:

Line 91: Remove the opening parenthesis before “TCID50”.

Line 170: Delete the word “mutant”.

Line 417: Replace “Promising” with “Potential.” Additionally, consider noting the risks of using different attenuated vaccines in the same region/country due to the potential for recombination.

Line 461: Remove the extra space.

Lines 519–520: Revise “This suggests that CD2v belongs to the pool of protective proteins.” to “This suggests that CD2v may belong to the pool of protective proteins.”

Lines 520–522: Indicate the reference number to support the statement: “Since protection induced by ASF MLV is seroimmunotype-specific.” (Pathogens 9(4):274; https://doi.org/10.3390/pathogens9040274).

Line 571: Insert a line break.

Lines 817–830: Insert line breaks between individual references for better readability.

Supplemental Table S1A: Add a footnote to explain the abbreviation “ON”.

Response 2: We appreciate your attention to details and have made all requested editorial changes:

Line 91 : The opening parenthesis before “TCIDâ‚…â‚€” has been removed.
Line 170 : The word “mutant” has been deleted as suggested.
Line 417 : The word “Promising” has been replaced with “Potential”.
Regarding the suggestion to discuss risks associated with using multiple attenuated vaccines in the same region due to possible recombination — we fully agree with its importance. However, we respectfully propose to elaborate on this issue in an upcoming comprehensive review article, where we plan to analyze mixed infections involving both virulent and vaccine strains, immunobiological characteristics of heterologous isolates, and related biosafety concerns in greater depth.
Line 461 : The extra space has been corrected.
Lines 519–520 : The sentence was modified from "This suggests that CD2v belongs to the pool of protective proteins" to "This suggests that CD2v may belong to the pool of protective proteins."
Lines 520–522 : We have added the appropriate citation [75] to support the statement: "Since protection induced by ASF MLV is seroimmunotype-specific..." .
Line 571 : A line break has been inserted.
Lines 817–830 : Line breaks between individual references have been added for improved readability.
Supplementary Table S1A : A footnote explaining the abbreviation "ON" (oronasal route) has been added to enhance clarity for readers.

Once again, we thank you for your valuable suggestions and for recognizing the improvements made in the current version of the manuscript. We believe these final edits further strengthen the quality and scientific value of the paper.

Should any additional changes be required, we will be happy to implement them promptly.